# A comparison of machine learning models versus clinical evaluation for mortality prediction in patients with sepsis

William P. T. M. van Doorn[1,2], Patricia M. Stassen[3,4], Hella F. Borggreve[3], Maaike J. Schalkwijk[3], Judith Stoffers[3], Otto Bekers[1,2], Steven J. R. Meex[1,2]*

1 Department of Clinical Chemistry, Central Diagnostic Laboratory, Maastricht University Medical Center, Maastricht, The Netherlands, 2 CARIM School for Cardiovascular Diseases, Maastricht University, Maastricht, The Netherlands, 3 Division of General Internal Medicine, Section Acute Medicine, Department of Internal Medicine, Maastricht University Medical Centre, Maastricht University, Maastricht, The Netherlands, 4 CAPHRI School for Care and Public Health Research Institute, Maastricht University, Maastricht, The Netherlands

* steven.meex@mumc.nl

**Data Availability Statement:** All relevant data are within the manuscript and its Supporting Information files.

## Abstract

### Introduction

Patients with sepsis who present to an emergency department (ED) have highly variable underlying disease severity, and can be categorized from low to high risk. Development of a risk stratification tool for these patients is important for appropriate triage and early treatment. The aim of this study was to develop machine learning models predicting 31-day mortality in patients presenting to the ED with sepsis and to compare these to internal medicine physicians and clinical risk scores.

### Methods

A single-center, retrospective cohort study was conducted amongst 1,344 emergency department patients fulfilling sepsis criteria. Laboratory and clinical data that was available in the first two hours of presentation from these patients were randomly partitioned into a development (n = 1,244) and validation dataset (n = 100). Machine learning models were trained and evaluated on the development dataset and compared to internal medicine physicians and risk scores in the independent validation dataset. The primary outcome was 31-day mortality.

### Results

A number of 1,344 patients were included of whom 174 (13.0%) died. Machine learning models trained with laboratory or a combination of laboratory + clinical data achieved an area-under-the ROC curve of 0.82 (95% CI: 0.80–0.84) and 0.84 (95% CI: 0.81–0.87) for predicting 31-day mortality, respectively. In the validation set, models outperformed internal medicine physicians and clinical risk scores in sensitivity (92% vs. 72% vs. 78%;p<0.001,all comparisons) while retaining comparable specificity (78% vs. 74% vs. 72%;p>0.02). The model had higher diagnostic accuracy with an area-under-the-ROC curve of 0.85 (95%CI:

**Funding:** This study was funded by a Noyons stipendium from the Dutch Federation of Clinical Chemistry (NVKC). The funders had no role in study design, data collection and analysis, decision to publish, or preparation of the manuscript.

**Competing interests:** The authors have declared that no competing interests exist.

0.78–0.92) compared to abbMEDS (0.63,0.54–0.73), mREMS (0.63,0.54–0.72) and internal medicine physicians (0.74,0.65–0.82).

## Conclusion

Machine learning models outperformed internal medicine physicians and clinical risk scores in predicting 31-day mortality. These models are a promising tool to aid in risk stratification of patients presenting to the ED with sepsis.

## Introduction

Among emergency department (ED) presentations, a substantial number of patients present with symptoms of sepsis [1]. Sepsis is defined as a systemic inflammatory response syndrome (SIRS) to an infection and is associated with a wide variety of risks including septic shock and death [2]. Mortality rates of sepsis are as high as 16%, potentially increasing up to 40% when suffering from septic shock [2, 3]. Novel clinical decision support (CDS) systems capable of identifying low- or high-risk patients could become important for early treatment and triage of ED patients, but also for preventing unnecessary referrals to the intensive care unit (ICU). EDs are one of the most overcrowded units of a modern hospital, highlighting the importance of proper allocation and management of resources [1]. Development of a risk stratification tool for patients with sepsis may improve health outcome in this group, but may also contribute to resolve the problem of overcrowded EDs.

Currently, a wide variety of clinical risk scores are used in routine clinical care to facilitate risk stratification of patients with sepsis [4]. These include the relatively simple (quick) sequential organ failure assessment ((q)SOFA) score [5, 6], but also more complex scores such as the abbreviated Mortality in Emergency Department Sepsis (abbMEDS) score and modified Rapid Emergency Medicine Score (mREMS) [7, 8]. These traditional risk scores have shown varying performance for predicting 28-day mortality (area under the receiver operating characteristic curve (AUC) for abbMEDS: 0.62–0.85, mREMS: 0.62–0.84 and SOFA: 0.61–0.82) [3, 8–11]. In addition, clinical judgment of the attending physician in the ED plays an important role in risk stratification. The judgment of physicians was found to be a moderate to good predictor (AUC of 0.68–0.81) of mortality in the ED [12, 13].

Interestingly, a new group of CDS systems are being developed based on machine learning (ML) technology [14]. Machine learning can extract information from complex, non-linear data and provide insights to support clinical decision making. Hence, the first studies emerged that report machine learning-based mortality prediction models using data from patients with sepsis presenting to the ED [15–26]. Unfortunately, these studies did not provide a comparison with physicians in terms of prognostic performance. Recently, a new group of machine learning algorithms termed gradient boosting trees emerged; showing superior performance compared to other ML models in some problems within the medical domain [27, 28]. Exploring if these models can outperform clinical risk scores and clinical judgment of physicians in their ability to identify low- or high-risk patients is a necessary step to explore the potential value of machine learning models in clinical practice.

The aim of this study was to develop machine learning-based prediction models for all-cause mortality at 31 days based on available laboratory and clinical data from patients presenting to the ED with sepsis. Subsequently, we compared the performance of these machine learning models with judgment of internal medicine physicians and clinical risk scores; abbMEDS, mREMS and SOFA.

## Methods

### Study design and setting

We performed a retrospective cohort study among all patients who presented to the ED at the Maastricht University Medical Centre+ between January 1, 2015 and December 31, 2016. All patients aged ≥18 years being referred to the internal medicine physician with sepsis, defined as a proven or suspected infection, and two or more SIRS and/or qSOFA criteria (S1 File) were included in this study [2, 5, 29]. Patients with missing clinical data or with less than four laboratory results were excluded. Also, patients who refused to give consent were excluded. This study was approved by the medical ethical committee (METC 2019–1044) and the hospital board of the Maastricht University Medical Centre+. Furthermore, the study follows the STROBE guidelines and was conducted according to the principles of the Declaration of Helsinki [30]. The ethics committee waived the requirement for informed consent.

### Data collection and processing

We collected clinical and laboratory data from all patients included in the study available within two hours after initial ED presentation. Clinical data were manually extracted through the electronic health record of the patient and included characteristics such as vital signs, hemodynamic parameters, and medical history (S1 Table). Biomarkers requested for standard clinical care were acquired through the laboratory information system. Biomarkers that were ordered in less than 1/1000 patients were excluded from the analysis. A list of included biomarkers is provided in S1 Table. Missing values did not require any processing as our machine learning model is capable of dealing with missing data. Instead, we created an additional variable for each biomarker with a discrete 'absence' or 'presence' feature to enable our model to distinguish between the absence and presence of a laboratory test within a patient. These features were included in both datasets. Finally, we derived two datasets from the processed data:

1. Laboratory dataset: this dataset consisted of age, sex, time of laboratory request and all requested laboratory biomarkers within two hours after the initial laboratory request

2. Laboratory + clinical dataset: this dataset contained all variables from the laboratory dataset, and additionally clinical, vital and physical (e.g. length and weight) characteristics of the patient

A full overview of all variables present in each dataset is described in S1 Table. Datasets were anonymized and randomly divided into two subsets: 1) a development subset (n = 1,244), used for model training and evaluation, and 2) an independent validation subset (n = 100), used for final validation and comparison of models with judgment of acute internal medicine physicians and clinical risk scores. A schematic overview of the study design and model development is depicted in Fig 1. Data processing and manipulation was performed using Python programming language (version 3.7.1) using packages numpy (version 1.17) and Pandas (version 0.24).

### Outcome measure

Septic shock during presentation was defined as systolic blood pressure (SBP) ≤90 mmHg and mean arterial pressure (MAP) ≤65 mmHg despite adequate fluid resuscitation. The outcome measure for this study was death within 31 days (1 month) after initial ED presentation. All-cause mortality information was acquired through electronic health records.

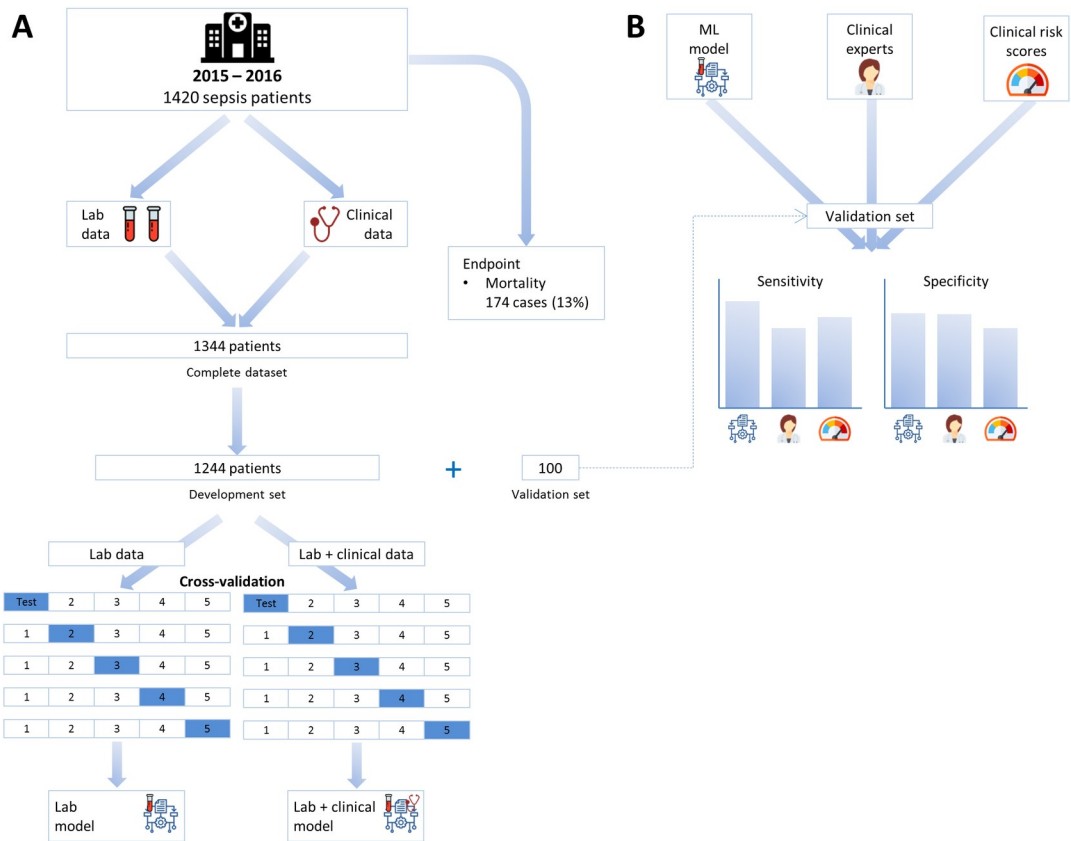

**Fig 1. Overview of study design and model development.** (A) We included 1,344 patients with a diagnosis of sepsis who presented to the ED. Patients were randomly partitioned in a development subset (n = 1,244), used to train and evaluate performance of machine learning models, and a validation subset (n = 100), used to compare models with internal medicine physicians and clinical risk scores. Cross-validation was used to obtain a robust estimate of model performance in the development subset. (B) The machine learning model with the highest cross-validation performance was compared internal medicine physicians and clinical risk scores to predict 31-days mortality.

## Model training and evaluation

Our proposed predictive model uses individual patient data available within two hours after initial ED presentation and generates the probability of mortality within 31 days. This prediction task can be solved by a variety of statistical and machine learning models. In the current study we evaluated logistic regression, random forest, multi-layer perceptron neural networks and XGBoost (S2 File and S2 Table) on the laboratory dataset. We selected XGBoost as our machine learning model of choice as this was proven to possess the highest baseline performance (S2 Table). XGBoost is a recent implementation of gradient tree boosting systems which involve combining the predictions of many "weak" decision trees into a strong predictor [27]. This recent implementation is characterized by integral support of missing data and regularization mechanisms to prevent overfitting [27]. XGBoost models and their development can be altered by adjusting the parameters of the technique, referred to as "hyperparameters". Due to sample size limitations and the scope of our study, we decided not to optimize our hyperparameters and predefined them as described in S3 Table.

We employed stratified K-fold cross validation to assess the generalizability of our prediction models. Briefly, we randomly partitioned the development subset (n = 1,244) into five, equally sized, folds. During each round of cross-validation, four of these folds were used to

train our models ("train set") and the fifth was used to evaluate performance ("test set"). This was done in such a manner that every fold would be labeled as test set only once. We monitored training and test set errors to ensure that training increased performance on the test set. Accordingly, training was terminated after 5,000 rounds or when performance on the test set did not further improve for 10 rounds. We evaluated developed models trained with (i) the laboratory dataset or (ii) the laboratory + clinical dataset, resulting in a total of two independent cross-validations.

## Model explanation

To explain the output of our XGBoost models, we used the SHapley Additive exPlanations (SHAP) algorithm, to help us understand how a single feature affects the output of the model [31–33]. SHAP uses a game theoretic approach to explain the output of any machine learning model. It connects optimal credit allocation with local explanations using the classic Shapley values from game theory and their related extensions [34, 35]. A Shapley value states, given the current set of variables, how much a variable in the context of its interaction with other variables contributes to the difference between the actual prediction and the mean prediction. That is, the mean prediction plus the sum of the Shapley values for all variables equals the actual prediction. It is important to understand that this is fundamentally different to direct variable effects known from e.g. (generalized) linear models. The SHAP value for a variable should not be seen as its direct -and isolated effect- but as its aggregated effect when interacting with other variables in the model. In our specific case, positive Shapley values contribute towards a positive prediction (death), whilst low or negative Shapely values contribute towards a negative prediction (survival). ML training and evaluation was done in Python using packages Keras (version 2.2.2), XGBoost (version 0.90), SHAP (version 0.34.0) and scikit-learn (version 0.22.1). The analysis code for this study is available on reasonable request.

## Comparison of machine learning with internal medicine physicians and clinical risk scores

Performance of machine learning models was compared with clinical judgment of acute internal medicine physicians (n = 4) and clinical risk scores in a validation subset of patients with sepsis (n = 100) which were not previously exposed to the ML model. We selected the best performing machine learning model from cross-validation and trained this with identical hyperparameters as previously described on the full development subset. A machine learning prediction of higher than 0.50 was considered as a positive prediction. Next, we calculated the mREMS, abbMEDS and SOFA clinical risk scores as described previously (S1 File) [8, 36]. Acute internal medicine physicians (n = 4; 2 experienced consultants in acute internal medicine and 2 experienced residents acute internal medicine) were asked to predict 31-day mortality in the validation subset, based on retrospectively collected clinical and laboratory data. This data was presented in the form of a simulated electronic health record.

## Statistical analysis

Descriptive analysis of baseline characteristics was performed using IBM SPSS Statistics for Windows (version 24.0). Continuous variables were reported as means with standard deviation (SD) or medians with interquartile ranges (IQRs) depending on the distribution of the data. Categorical variables were reported as proportions. Cross-validated models were assessed by receiver operating characteristic (ROC) curves and compared by their AUC using the Wilcoxon matched-pairs signed rank test. Besides diagnostic performance, we assessed calibration in cross-validations with reliability curves [37] and brier scores [38]. In our final validation

subset, we compared the predictive performance of our best performing ML model to the judgment of acute internal medicine physicians and clinical risk scores with respect to sensitivity, specificity, positive predictive value (PPV), negative predictive value (NPV), accuracy and AUC. Differences in AUC were tested using the method of DeLong et al [39]. Confidence intervals for proportions (e.g. sensitivity) were calculated using binomial testing and compared using McNemar's test. To analyze individual differences between internal medicine physicians, we performed two additional sensitivity analyses. First, the Cohen κ statistic was used to measure the inter-observer agreement between the internal medicine physicians. The level of agreement was interpreted as nil if κ was 0 to 0.20; minimal, 0.21 to 0.39; weak, 0.40 to 0.59; moderate, 0.60 to 0.79; strong, 0.80 to 0.90; and almost perfect, 0.90 to 1 [40]. Second, we compared the machine learning model against alternating groups of internal medicine physicians in which one physician was removed in each comparison.

## Results

### Study population and characteristics

During the study period, 5,967 patients presented to the ED who were referred to an internal medicine physician in our hospital. Of these patients, we included 1,420 patients with a suspected or proven infection, fulfilling the SIRS and/or qSOFA criteria. A number of 76 patients were excluded due to missing clinical data (n = 23) and insufficient number of laboratory results (n = 53), to form a final cohort of 1,344 patients (S1 Fig). Among all patients, 102 (7.6%) suffered from septic shock during presentation at ED and 174 (13.0%) died within 31 days after initial ED presentation. Baseline characteristics of the study patients in development and validation datasets are shown in Table 1.

### Machine learning development and evaluation

To assess the generalizability of our developed XGBoost models, we employed five-fold cross validation on the development dataset (n = 1,244). XGBoost models trained with laboratory data achieved an AUC of 0.82 (95% CI: 0.80–0.84) for predicting 31-day mortality (Fig 2). The performance improved, although not statistically significant, when clinical data was added to the laboratory data to train XGBoost to an AUC of 0.84 (95% CI: 0.81–0.87) for predicting mortality (compared to lab only; p = 0.25). Individual cross-validation results of each model are depicted in S2 Fig. Calibration curves show well calibrated models with brier scores between 0.08 to 0.10 (S3 Fig).

### Model explanation

To identify which laboratory and clinical features contributed most to the performance of our models, we calculated SHAP values for the (i) laboratory and (ii) laboratory + clinical models (Fig 3). Among the highest ranked features, we observe features that are also often used in risk scores including urea, platelet count, glasgow coma score (GCS) and blood pressure. Interestingly, we also observe features such as glucose, lipase, and GCS which are less commonly associated with mortality in sepsis patients. An extended analysis of the correlation between important features in our models and risk scores is provided in S4 Table. Moreover, these SHAP plots allow us to examine the individual impact of laboratory and clinical features on the predictions of our models. For example, higher urea and C-reactive protein (CRP) levels (represented by red points) have a high SHAP value and thus a positive effect on the model outcome (death).

**Table 1. Baseline characteristics of patients in the development and validation datasets.**

| Characteristics | Development N = 1,244 | Validation N = 100 |
|---|---|---|
| Demographics | | |
| Age | 71.3 (58.8–82.3) | 70.8 (58.4–82.8) |
| Sex, female | 567 (45.6) | 58 (58.0) |
| Comorbidity | | |
| Cancer | 446 (35.9) | 28 (28.0) |
| Cardiopulmonary | 381 (30.6) | 30 (30.0) |
| Diabetes | 264 (21.2) | 19 (19.0) |
| Renal disease | 128 (10.3) | 9 (9.0) |
| Liver disease | 42 (3.4) | 7 (7.0) |
| Neuropsychiatric | 65 (5.2) | 2 (2.0) |
| Focus of infection at ED | | |
| Respiratory tract | 421 (33.8) | 34 (34.0) |
| Urinary tract | 218 (17.5) | 18 (18.0) |
| Gastrointestinal tract | 415 (33.4) | 37 (37.0) |
| Others | 75 (6.0) | 6 (6.0) |
| Skin | 115 (9.2) | 5 (5.0) |
| Severity scores | | |
| abbMEDS[a] | 5.5 (3–8) | 6 (3–8) |
| mREMS[b] | 7 (6–9) | 7 (6–9) |
| SOFA[c] | 7 (5–9) | 6 (5–8) |
| Outcomes | | |
| Septic shock | 94 (7.6) | 8 (8.0) |
| 31-day mortality | 161 (12.9) | 13 (13.0) |

[a] AbbMEDS, Abbreviated Mortality in ED Sepsis, was calculated as described by Vorwerk et al [8].

[b] mREMS, modified Rapid Emergency Medicine Score, was calculated as described by Chang et al [36].

[c] SOFA, Sepsis-related Organ Failure Assessment, was calculated as described by Vincent et al [6].

## Machine learning versus internal medicine physicians and clinical risk scores

To explore the potential value of machine learning models in clinical practice, we compared the model trained with laboratory + clinical data with acute internal medicine physicians and clinical risk scores, abbMEDS, mREMS and SOFA, to predict 31-day mortality. In an independent validation subset (n = 100) -which the model never had been exposed to before- it achieved a sensitivity of 0.92 (95% CI: 0.87–0.95, Fig 4A) and specificity of 0.78 (95% CI: 0.70–0.86, Fig 4B). In terms of sensitivity, the machine learning model significantly outperformed internal medicine physicians (0.72, 95% CI: 0.62–0.81; p<0.001), abbMEDS (0.54, 95% CI: 0.44–0.64; p<0.0001), mREMS (0.62, 95% CI: 0.52–0.72; p<0.001) and SOFA (0.77, 95% CI: 0.69–0.85; p = 0.003). On the other hand, the model retained a specificity that was comparable to that of internal medicine physicians (0.74, 95% CI: 0.64–0.82; p = 0.509), abbMEDS (0.72, 95% CI: 0.64–0.81; p = 0.327) and SOFA (0.74, 95% CI: 0.65–0.82, p = 0.447), while still outperforming mREMS (0.64, 95% CI: 0.55–0.74; p = 0.02). Additionally, the model had higher overall diagnostic accuracy with an AUC of 0.852 (95% CI: 0.783–0.922) compared to abbMEDS (0.631, 0.537–0.726, p = 0.021), mREMS (0.630, 0.535–0.724, p = 0.016), SOFA (0.752, 0.667–0.836, p = 0.042) and internal medicine physicians (0.735, 0.648–0.821, p = 0.032–0.189) (S4 Fig and S5 Table). Similar observations were made in additional evaluation metrics

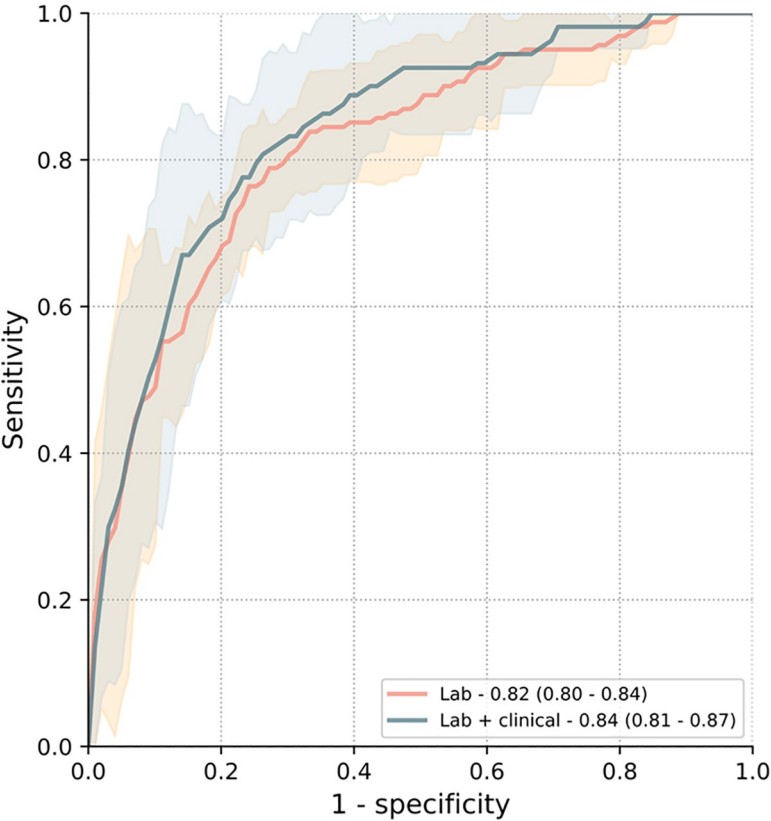

**Fig 2. XGBoost model performance for predicting all-cause mortality at 31 days in the development dataset.**
Models trained with laboratory data achieved a mean AUC of 0.82 (95% CI: 0.80–0.84) for predicting 31-day mortality. Predictive performance increased when models were trained with laboratory + clinical data to a mean AUC of 0.84 (95% CI: 0.81–0.87), but this was not statistically different (p = 0.25).

such as positive predictive value (NPV), negative predictive value (NPV) and accuracy (S5 Table). Individually, consultants were found to be more sensitive compared to residents (S5 Fig) with a poor to moderate agreement between the internists (Cohen's Kappa 0.46 to 0.67) (S6 Table). A sensitivity analysis with four additional comparisons, where one physician was excluded at a time, confirmed that the results are robust and that the outperformance of the machine learning model was not due to an outlier in the physician group (S7 Table).

## Discussion

In the present study we demonstrate the application of machine learning models to predict 31-day mortality patients presenting to the ED with sepsis. Our study reports several important findings.

First, we show that machine learning based models can accurately predict 31-day mortality in patients with sepsis. Highest diagnostic accuracy was obtained with the model that was trained with both laboratory and clinical data. Patient characteristics that are employed in traditional risk scores, such as blood pressure and heart rate, were also found to be amongst the most important variables for model predictions. Second, machine learning models outperformed the judgment of internal medicine physicians and commonly used clinical risk scores, abbMEDS, mREMS and SOFA. Specifically, machine learning was more sensitive compared with risk scores and internal medicine physicians, while retaining identical or slightly higher

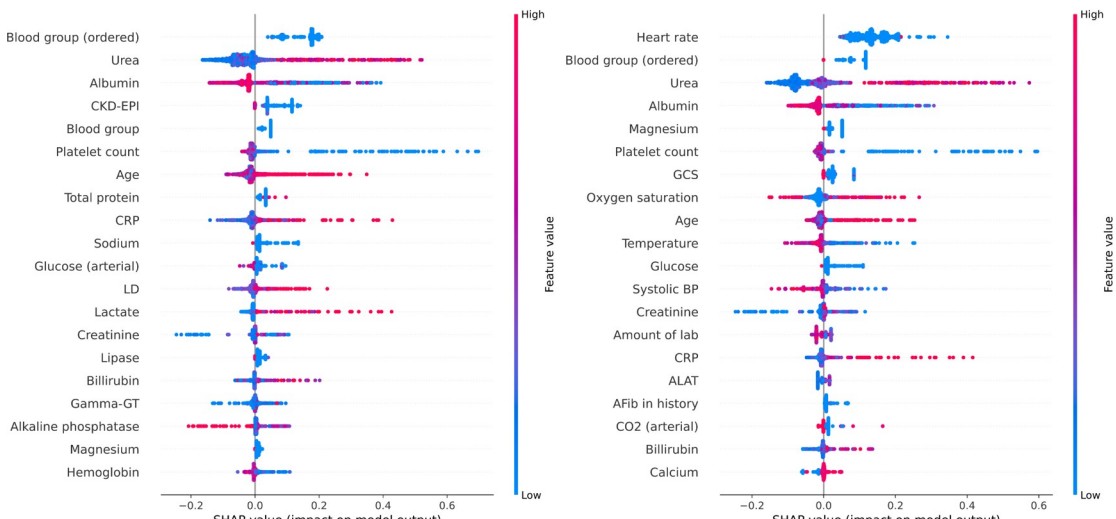

**Fig 3. Analysis of parameter importance in the XGBoost models.** Models with laboratory data (left) and with laboratory + clinical data (right) were analyzed using SHAP values. Individual parameters are ranked by importance in descending order based on the sum of the SHAP values over all the samples. Negative or low SHAP values contribute towards a negative model outcome (survival), whereas high SHAP values contribute towards a positive model outcome (death).

specificity. These preliminary data provide support in favor of the development and implementation of machine learning based models as clinical decision support tools, e.g. risk stratification of sepsis patients presenting to the ED.

We are aware of several studies which describe the machine-learning based prediction of mortality in sepsis populations presenting to the ED [15–17]. Taylor et al. described a random forest model outperforming clinical risk scores in an ED population. Despite their bigger population, our XGBoost model appears to achieve similar performance to their random forest model, which corroborates and extends the power of this machine learning technique. Two recent studies by Barnaby et al. and Chiew et al. focused on using heart rate variability (HRV) for risk prediction in sepsis patients and reported predictive performance similar to our findings [15, 16]. Interestingly, their populations were smaller and this would therefore also advocate the use of HRV in our models. Despite these findings, Chiew et al. demonstrated that models without laboratory data significantly decreased in performance, emphasizing the importance of laboratory data in these machine learning models. Nevertheless, to the best of our knowledge this is the first study to report the direct comparison of machine learning models with internal medicine physicians. Although we do not present prospective results, we demonstrate that machine learning outperforms clinical judgment of internal medicine physicians and clinical risk scores, implying that current XGBoost models potentially aid in risk stratification of ED patients. As an example, implementation of these models should revolve around identifying patients with a high risk, e.g. ≥50% mortality within 31 days, which would then be re-evaluated once more before being discharged from the ED. This kind of implementation was shown in a recent randomized clinical trial by Shimabukuro et al. [41], proving that average length of stay and in-hospital mortality decreased by using a ML-based sepsis detection model in the ICU. Although this was carried out with a small population in an ICU instead of the ED, it clearly shows the potential of ML-based risk stratifying models.

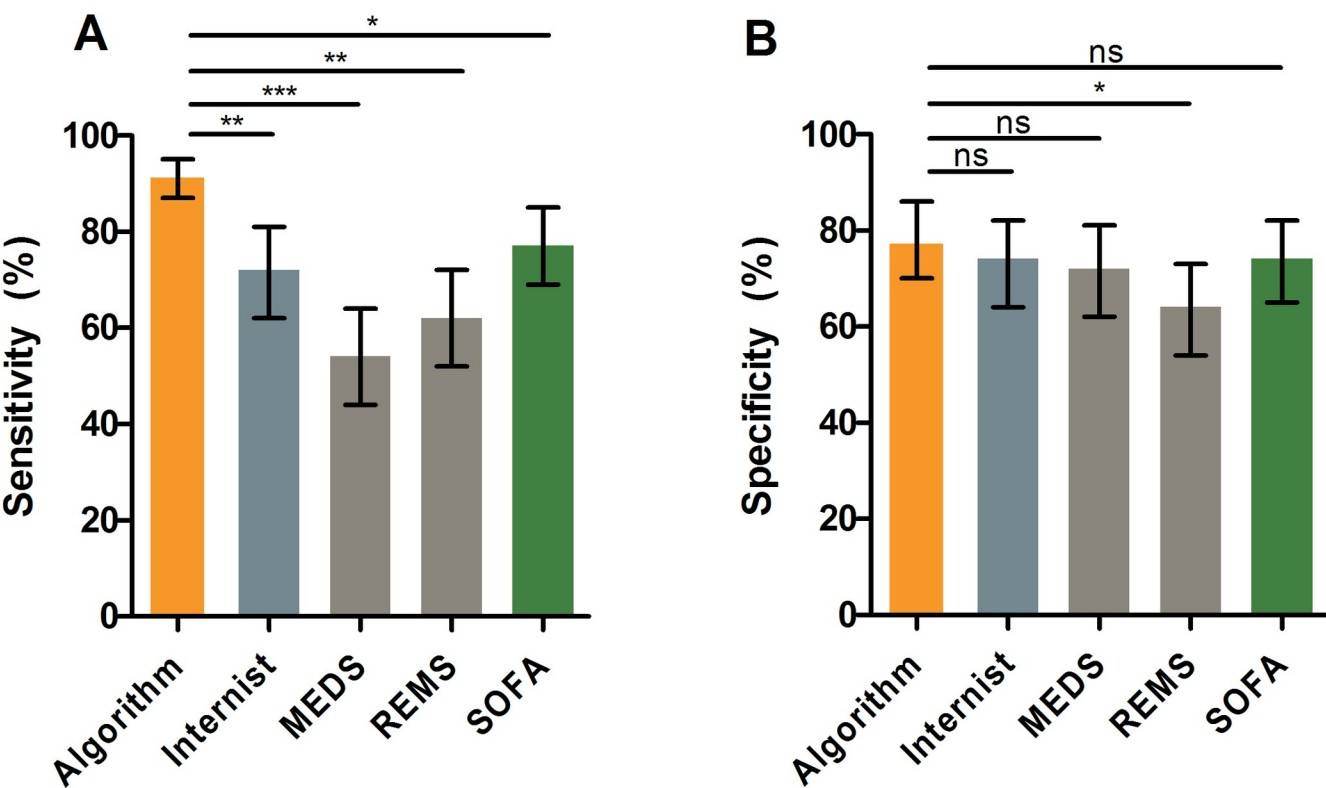

**Fig 4. Comparison of XGBoost model with internal medicine physicians and clinical risk scores.** The XGBoost model achieved a sensitivity (A) of 0.92 (95% CI: 0.87–0.95) and specificity (B) of 0.78 (95% CI: 0.70–0.86) for predicting mortality. This was significantly better than the mean prediction of internal medicine physicians for sensitivity (0.72, 0.62–0.81; p<0.001) as well as abbMEDS (0.54, 0.44–0.64; p<0.0001), mREMS (0.62, 0.52–0.72; p<0.001) and SOFA (0.77, 95% CI: 0.69–0.85; p = 0.003). In terms of specificity, internal medicine physicians (0.74, 0.64–0.82; p = 0.509), abbMEDS (0.72, 0.64–0.81; p = 0.327) and SOFA (0.74, 95% CI: 0.65–0.82, p = 0.447) achieved similar performance compared to the XGBoost model, opposed to mREMS (0.64, 0.55–0.74; p = 0.02) which was significantly worse than machine learning predictions. * = p<0.05; ** = p<0.001; *** = p<0.0001; NS = not significant.

The current study has several strengths and limitations. Strengths include (i) comparison of laboratory versus laboratory + clinical models, (ii) analysis of features contributing to models' prediction and (iii) the comparison with internal medicine specialists. We are also aware of several limitations. First, the present study was a single-center study with a relatively small sample size at least from a machine learning analysis perspective. Nearly all machine learning models scale exceptionally well with data, and therefore substantial further improvement of diagnostic accuracy is likely when increasing the sample size. We also limited ourselves to sepsis patients presenting to the ED, and thus it is unknown to what degree these models translate to a broader, general ED population. Second, results presented in this study are based on retrospective data in a single center, limiting the external validity of the model. Unfortunately, this limitation currently applies to most studies applying ML in medicine. Third, the present study focused on model development and subsequent performance comparison with clinical judgment and clinical risk scores. It should be noted that the comparison with internal medicine specialists was performed using retrospectively generated electronic health records, rather than a prospective evaluation, which might have underestimated their diagnostic performance as they were not able to directly "see" the patient. Prospective evaluation, in respect to mortality, but also in relation to clinical endpoints that confirm true clinical benefit would facilitate implementation of ML-based risk stratification tools in clinical practice.

## Conclusion

In conclusion, the present proof-of-concept study demonstrates the potential of machine learning models to predict mortality in patients with sepsis presenting to the ED. Machine learning outperformed clinical judgment of internal medicine physicians and established clinical risk scores. These data provide support in favor of the implementation of machine learning based risk stratification tools of sepsis patients presenting to the ED.

## Supporting information

**S1 File. Extended description of clinical criteria and risk scores.**
(DOCX)

**S2 File. Background information on machine learning models reviewed in the current study.**
(DOCX)

**S1 Table. Overview of variables present in the datasets.** The laboratory dataset consisted exclusively of laboratory variables with age, sex and time of request. The laboratory and clinical dataset contained all variables from the laboratory dataset and additionally clinical and vital characteristics.
(DOCX)

**S2 Table. Comparison of baseline statistical and machine learning models for predicting 31-day mortality risk.** We performed a baseline comparison of statistical and machine learning models (S1 File) for the 31-day mortality prediction task using the laboratory dataset. We used five-fold cross validation to assess model performance. Performance was assessed by area under the receiver operating characteristic curve (AUC) and accuracy. Confidence intervals were calculated using bootstrapping methods (n = 1,000).
(DOCX)

**S3 Table. Hyperparameters of XGBoost models.** Hyperparameters were based on theoretical reasoning rather than hyperparameter tuning. This was done to prevent overfitting on hyperparameters due to small sample size. "Base_score", "Missing", "Reg_alpha", "Reg_lambda" and "Subsample" parameters were standard values provided by the XGBoost interface. "Max_depth", "max_delta_step" and "estimators" were values we internally use for these kind of machine learning models. During the study, hyperparameters were never adjusted to gain performance in our validation dataset.
(DOCX)

**S4 Table. Extended analysis of correlation between important model features and clinical risk scores.** To study the correlation between the most important features contributing to model predictions and the clinical criteria (qSOFA and SIRS) and risk scores (abbMEDS and mREMS), we compared their existence in both. The top-20 most important features (Fig 3 in main article) are compared to all criteria in the clinical scores (S1 File). We observe that most of the features present in the clinical criteria and scores are also among the most important features in the lab and clinical machine learning model.
(DOCX)

**S5 Table. Extended comparison of machine learning models with internal medicine physicians and clinical risk scores.** In addition to sensitivity and specificity, we evaluated the performance of each group by positive predictive value (PPV), negative predictive value (NPV), accuracy and area-under-the receiver operating characteristics curve (AUC). Our XGBoost

model shows superior performance in each of these metrics, which is in line with the findings presented in the manuscript.
(DOCX)

**S6 Table. Inter-rater agreement of internal medicine physicians.** Cohen's kappa was used to measure the inter-rater agreement between the internal medicine physicians. The level of agreement was interpreted as nil if κ was 0 to 0.20; minimal, 0.21 to 0.39; weak, 0.40 to 0.59; moderate, 0.60 to 0.79; strong, 0.80 to 0.90; and almost perfect, 0.90 to 1.3.
(DOCX)

**S7 Table. Machine learning comparison to alternating physician groups.** In each comparison between the machine learning model and the physicians group, a single physician was removed from the physician group. In every comparison the machine learning model outperforms the physicians. This analysis shows that the higher performance of the machine learning model was not due to systemic underperformance of a single physician.
(DOCX)

**S1 Fig. Flow diagram of study inclusion.** During the study period 5,967 patients that presented to our emergency department were referred to an internal medicine physician. Of these patients, 1420 patients fulfilled two or more SIRS and/or qSOFA criteria. After exclusion of 76 patients, a number of 1,344 patients were separated into development and validation datasets.
(DOCX)

**S2 Fig. Five-fold cross validation of diagnostic performance of XGBoost models.** During each cycle of cross-validation, we assessed predictive performance by area under the receiver operating characteristic curves (AUC). Performance was determined for models trained with laboratory data (A) and models trained with laboratory and clinical data (B) to predict 31-day mortality.
(DOCX)

**S3 Fig. Five-fold cross validation of calibration of XGBoost models.** During each cycle of cross-validation, we assessed calibration by calibration curves and their respective brier scores. Calibration was determined for models trained with laboratory data (A) and models trained with laboratory and clinical data (B).
(DOCX)

**S4 Fig. Receiver operating characteristic analysis of machine learning model, risk scores and internal medicine physicians.** Receiver operating characteristics analysis of the lab + clinical machine learning model (AUC: 0.852 [0.783–0.922]), abbMEDS (0.631 [0.537–0.726]), mREMS (0.630 [0.535–0.724]) and internal medicine physicians (mean 0.735 [0.648–0.821]). Internal medicine physicians were depicted as bullets in the ROC analysis.
(DOCX)

**S5 Fig. Individual performance of internal medicine physicians.** Predictive performance of all internal medicine specialists (n = 4; 2 experienced consultants in acute internal medicine and 2 experienced residents acute internal medicine) was assessed by sensitivity (left) and specificity (right). Consultants (experienced) specialists are depicted in grey and residents in orange.
(DOCX)

## Author Contributions

**Conceptualization:** William P. T. M. van Doorn, Steven J. R. Meex.

**Data curation:** William P. T. M. van Doorn, Patricia M. Stassen, Hella F. Borggreve, Maaike J. Schalkwijk, Judith Stoffers, Otto Bekers, Steven J. R. Meex.

**Formal analysis:** William P. T. M. van Doorn, Patricia M. Stassen.

**Funding acquisition:** William P. T. M. van Doorn, Otto Bekers.

**Investigation:** William P. T. M. van Doorn, Patricia M. Stassen, Hella F. Borggreve, Maaike J. Schalkwijk, Judith Stoffers, Otto Bekers, Steven J. R. Meex.

**Methodology:** William P. T. M. van Doorn, Steven J. R. Meex.

**Project administration:** William P. T. M. van Doorn.

**Resources:** William P. T. M. van Doorn, Hella F. Borggreve, Otto Bekers, Steven J. R. Meex.

**Software:** William P. T. M. van Doorn.

**Supervision:** Otto Bekers, Steven J. R. Meex.

**Validation:** William P. T. M. van Doorn, Steven J. R. Meex.

**Visualization:** William P. T. M. van Doorn, Steven J. R. Meex.

**Writing – original draft:** William P. T. M. van Doorn, Patricia M. Stassen, Hella F. Borggreve, Maaike J. Schalkwijk, Judith Stoffers, Otto Bekers, Steven J. R. Meex.

**Writing – review & editing:** William P. T. M. van Doorn, Patricia M. Stassen, Hella F. Borggreve, Maaike J. Schalkwijk, Judith Stoffers, Otto Bekers, Steven J. R. Meex.

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
