## [Decision Letter · Decision Letter 0]

23 Jun 2020

PONE-D-20-09068

Machine Learning versus Physicians to Predict Mortality in Sepsis Patients Presenting to the Emergency Department

PLOS ONE

Dear Dr. Meex,

Thank you for submitting your manuscript to PLOS ONE. After careful consideration, we feel that it has merit but does not fully meet PLOS ONE’s publication criteria as it currently stands. Therefore, we invite you to submit a revised version of the manuscript that addresses the points raised during the review process.

We look forward to receiving your revised manuscript.

Kind regards,

Ivan Olier, Ph.D.

Academic Editor

PLOS ONE

Journal Requirements:

2. In your ethics statement in the manuscript and in the online submission form, please provide additional information about the patient records used in your retrospective study. Specifically, please ensure that you have discussed whether all data were fully anonymized before you accessed them and/or whether the IRB or ethics committee waived the requirement for informed consent. If patients provided informed written consent to have data from their medical records used in research, please include this information.

3. Please provide additional details regarding participant consent. In the ethics statement in the Methods and online submission information, please ensure that you have specified (1) whether consent was informed and (2) what type you obtained (for instance, written or verbal, and if verbal, how it was documented and witnessed)

5. Please ensure that you refer to Figure 4 in your text as, if accepted, production will need this reference to link the reader to the figure.

Reviewers' comments:

Reviewer's Responses to Questions

**Comments to the Author**

1. Is the manuscript technically sound, and do the data support the conclusions?

Reviewer #1: Partly

Reviewer #2: Partly

2. Has the statistical analysis been performed appropriately and rigorously? 

Reviewer #1: Yes

Reviewer #2: No

3. Have the authors made all data underlying the findings in their manuscript fully available?

Reviewer #1: Yes

Reviewer #2: No

4. Is the manuscript presented in an intelligible fashion and written in standard English?

Reviewer #1: Yes

Reviewer #2: Yes

5. Review Comments to the Author

Reviewer #1: The manuscript presents the development of a machine learned model predicting 31-day mortality.

The predictive performance of the model is compared to the predictive performance of 4 internists (2 consultants and 2 fellows). The machine learned model outperformed the internists by a wide margin.

The novelty in the manuscript is the comparison between the machine learning (ML) model and the internists.

Developing yet another ML model for sepsis mortality risk is not exciting; but a comparison of the model with

human expert internists is exciting. As the key contribution of the paper this comparison has to be solid.

Unfortunately, the way the paper stands now, I do not feel that this comparison is rock solid. Here are some

weakness in the comparison.

(a) Although the difference between internist and ML is high, with only 4 internists, the differences among the internists can significantly influence the results. It would be helpful to have inter-rater agreement among the internists.

Also, how was specificity/sensitivity for the ML model determined?

(b) It should also be noted that the machine learned model incorporates physician judgement. Whether a lab test is performed or not is based on clinical judgement and is made available to the model (as the lab absence/presence indicator). A person with fewer labs have fewer problems and is hence less likely to die. How the authors handle missing values is typically completely reasonable and correct, however, in this case, it "leaks" physician knowledge to the ML model.

Beside the comparison, the model development itself raises some concerns. Specifically, he performance differences between the various machine learning methods is incredibly high (.63-ish for Ridge regression, .65 neural networks, .72 for random forest, but .85 for xgboost). We also typically see xgboost outperform other methods but not to this extreme extent. I am wondering whether the authors may have made some mistake evaluating xgboost. There are other indicators of a possible mistake:

- The performance of xgboost differs across different tables (.813 in Supp Tbl 2; .852 in Supp Tbl 5). With a stated confidence interval of .79-.83, .852 is quite a bit outside the confidence interval.

- The learning rate is stated as .075 in Supp Tbl 5 but .001 in the text; could be a typo or actual different hyper-parameterization. A different hyper-parameterization could explain observed AUC values outside the confidence interval than random chance.

- The "95%"-confidence intervals are not 95%. In Supp Fig. 2B, only fold 1 falls consistently within the stated 95% confidence interval, the remaining 4 folds fall outside the "95%"-confidence interval for large consecutive portions of the ROC curve. How was the bootstrap estimation performed? Specifically, which data set is resampled (the development or the leave-out)?

I agree with the authors that a full-fledged lattice search of the hyper-parameter space is unnecessary, but I would suggest:

- explaining how the hyper-parameters were determined (e.g. using the default values from the package)

- explaining which if any hyper-parameters were changed (based on the CV test set; NOT the 100 patient leave-out set)

- conduct a sensitivity analysis demonstrating that small changes in the hyper parameters do not lead to major perturbations in the performance.

The SHAP score is poorly explained and Figure 3 is poorly described. Readers not familiar with SHAP and violin plots will not understand this figure. An example could be useful. Eg. High values of urea (red points) impact the risk of mortality positively (large positive impact); low values reduce it slightly (modest negative impact); urea observations in most patients have no impact on mortality (the urea violin plot is widest at 0 impact).

A concern with xgboost is the (somewhat) black-box nature of the model. The authors address this concern by computing the SHAP score for each feature. Given that the key differentiator of trees from the other methods is their ability to detect and use interactions, SHAP may not capture this well. A better approach is to take the (say) top 20 features and build a reduced model to see the performance loss. (I picked 20 because that is what the authors focus on in Supp Tbl 4, but other numbers could illustrate their point better.)

Clinical significance.

ML models in sepsis abound with very marginal contributions. Proposing yet another one without explaining how it can improve sepsis care is fairly meaningless. I appreciate the authors mentioning how their model could be used, and I think expanding on this is essential. The author's suggestion of using the model to identifying high risk patients (>= 50% risk of mortality) is reasonable. However, their evaluation does not quantify their contribution from this perspective. The clinical significance of this work could be significantly improved by (i) showing how much better the ML model is at identifying patients at >=50% risk (or any other high risk); (ii) clinically describing where the ML model was correct and the internists were not (again >= 50% vs <50% can be used) ; (iii) understanding the limits of the model, clinically describing where the ML model "fails": predicts low risk of mortality while the patient dies (regardless of whether the internists made the correct prediction). Note that higher AUC does not guarantee an improved ability to identify high-risk patients; correctly ordering low-risk patients will increase the AUC yet it has not clinical significance.

Limitations. Some important limitation are omitted.

1.The key limitation is portability. We can expect the risk scores and the internists to have similar performance if they made predictions for patients in a different health system. How about the xgboost model? Will it achieve similar performance?

2. Physicians, which actually seeing patients, may incorporate other features not captured by the EHR.

SUMMARY.

The key contribution of the paper is a comparison between an ML model and human internists. There are minor flaws in the comparison (ML has access to clinical judgement).

The model development process appears to have technical problems (confidence intervals appear incorrect; unknown rational for the hyper-parameterization and dependence of the performance on these parameters).

The performance envelope of the model is unexplored: what clinical patient characteristics make it fail? make it perform better than the internists?

Important limitations are not mentioned.

I think this paper has the potential to be an influential piece of work but its contribution as it stands now is insufficient and some technical aspects may even be incorrect.

Reviewer #2: • Mortality prediction in patients with sepsis, utilising a small dataset of patients presenting to a single emergency department

• The title is a little sensational – “Machine Learning versus Physicians”, and could be modified to represent the science

o “A comparison of machine learning models versus clinical evaluation for mortality prediction in patients with sepsis”

• Authors testing the hypotehesis that machine learning models would outperform physician evaluation and existing clinical risk scores

• Authors determined that machine learning models out performed clinicians due to higher sensitivity and specificity, however discriminatory information is not provided in the abstract

• Introduction

o The references noted in the introduction note high discriminatory scores for abbMEDS (up to 0.85), mREMS (up to 0.84) and clinician judgement (up to 0.81), suggesting similar performance to this cohort

o The comment is made that “machine learning can extract information from incomplete…data” – although complexity and non-linear relationships are areas where ML does succeed, incomplete data is still a major limitation

• Methods

o Patients with missing clinical data were excluded – how many variables needed to be missing? Were the data missing at random? A plot representing percentage of missing values would be valuable, and an analysis to confirm that the missing value distribution was equal in the training and testing set

o Similarly, the authors note that the machine learning model is capable of dealing with missing data – could further information be provided as to how GBMs deal with the issue? Further, this statement is incongruent with the previous noting that patients with missing clinical data were excluded (unless this does not refer to biomarkers, and instead to other data)

o As biochemical markers were taken within the first two hours, and a significant proportion of treatment is performed in that time frame, was the variable of time between presentation and time of biomarker withdrawal recorded and adjusted for?

o The train/test split ratio is unusual – 93:7 – can the authors explain why this was chosen? Internal validation on 100 patients is small and may affect the reproducibility of this result.

o Mortality was obtained from the electronic health record – was this linked to a national health index? How were deaths out of hospital recorded in the EHR?

o The septic shock definition mentions a MAP ≤90 – is an alternate threshold intended here?

o The use of SHAP to explain the findings are an important addition and the authors should be credited for its inclusion

o Why were internal medicine physicians chosen, as opposed to emergency or intensive care physicians? Is this typical for this hospital?

o Calibration measures such as the Brier score and calibration plots should be presented for the models

o There is marked class imbalance, with only 13% of the population experiencing the primary outcome of death – were oversampling methods considered?

• Results and Discussion

o There appears to be a significantly higher percentage of patients with cancer and diabetes in the training/development set

o Why is the discrimination/AUC not directly compared between the ML models and the physicians and clinical scores?

o AUCs should be compared using DeLong’s test to demonstrate statistical superiority in regard to discrimination

o Figure 2 does not include the clinical risk scores for comparison

o Creatinine does not share the same relationship with model output as does urea – can this be explained clinically? (from the SHAP figure)

o Several references to other machine learning models in the literature focused on sepsis and mortality prediction have been omitted

General points

• Funded study with no conflicts of interest

• Grammar could be slightly improved however does not interfere with message of the paper

o “…and categorize from low to high risk” should be “and is categorized from low to high risk”

o “follow-studies” should be “follow-up studies”

• Authors have made data available without restriction, however note that it is available in the supplementary material – the code and raw data are not provided?

6. PLOS authors have the option to publish the peer review history of their article (what does this mean?). If published, this will include your full peer review and any attached files.

Reviewer #1: No

Reviewer #2: No

---

## [Author Response · Author response to Decision Letter 0]

10 Jul 2020

Dr. Ivan Olier, PhD

Academic Editor of PLOS ONE

Faculty of Engineering and Technology

Liverpool John Moores University, United Kingdom

July 10, 2020

Dear Dr. Olier,

We wish to thank you for the interest in our manuscript. We appreciate the constructive comments of both reviewers which have been valuable for improving of our manuscript. Please find enclosed our revision of the manuscript “A comparison of machine learning models versus clinical evaluation for mortality prediction in patients with sepsis” (title has changed). We addressed the comments of both reviewers in a point by point rebuttal and revised our manuscript, accordingly. Track changes were used to indicate the revised sections in our manuscript. Additionally, minor textual changes were performed to improve the readability of the manuscript.

On behalf of my co-authors, I am pleased to submit a revised version of our manuscript. I confirm that this work is original and has not been published elsewhere nor is it currently under consideration for publication elsewhere. All authors have read and agreed with the revision of the manuscript. 

We wish to thank you for the opportunity to submit a revised version of our manuscript to PLOS ONE.

Yours sincerely,

Steven J.R. Meex, Ph.D.

Central Diagnostic Laboratory

Maastricht University Medical Center+

Post office box 5800

6202 AZ Maastricht

The Netherlands

Tel: +31 (0)43-387 4709

Fax: +31 (0)84-003 8525

E-mail: steven.meex@mumc.nl

Journal Requirements:

1.Please ensure that your manuscript meets PLOS ONE's style requirements, including those for file naming. The PLOS ONE style templates can be found at https://journals.plos.org/plosone/s/file?id=wjVg/PLOSOne_formatting_sample_main_body.pdf andhttps://journals.plos.org/plosone/s/file?id=ba62/PLOSOne_formatting_sample_title_authors_affiliations.pdf

We adjusted the manuscript and supplemental information to meet the style requirements of PLOS ONE.

2. In your ethics statement in the manuscript and in the online submission form, please provide additional information about the patient records used in your retrospective study. Specifically, please ensure that you have discussed whether all data were fully anonymized before you accessed them and/or whether the IRB or ethics committee waived the requirement for informed consent. If patients provided informed written consent to have data from their medical records used in research, please include this information.

The data was fully anonymized before analysis. The study was approved by the medical ethical committee (METC 2019-1044) and hospital board of the Maastricht University Medical Centre+. The ethics committee waived the requirement for informed consent. We adjusted our ethics statement and methods sections in the manuscript to include this necessary information (page 7, lines 122-123).

3. Please provide additional details regarding participant consent. In the ethics statement in the Methods and online submission information, please ensure that you have specified (1) whether consent was informed and (2) what type you obtained (for instance, written or verbal, and if verbal, how it was documented and witnessed)

The ethics committee waived the requirement for informed consent and therefore we did not collect this. We adjusted our ethics statement and methods sections in the manuscript to include this necessary information (page 7, lines 122-123).

We added the ORCID iD for the corresponding author in the editorial manager.

5. Please ensure that you refer to Figure 4 in your text as, if accepted, production will need this reference to link the reader to the figure.

We inserted two references to Figure 4 in the last section of our results section (page 20, line 328).

We inserted the captions for supporting information at the end of our manuscript file and also adjusted the citations accordingly.

Reviewer 1

We are pleased with the feedback given by the reviewer. We want to thank the reviewer for evaluating our manuscript and for giving important suggestions and comments to improve our manuscript. The comments have been addressed in a point-by-point fashion in this document. The changes are highlighted in track changes throughout the manuscript. The pages and lines mentioned for each adjustment refer to the manuscript and supplemental file with track changes.

1. The manuscript presents the development of a machine learned model predicting 31-day mortality. The predictive performance of the model is compared to the predictive performance of 4 internists (2 consultants and 2 fellows). The machine learned model outperformed the internists by a wide margin. The novelty in the manuscript is the comparison between the machine learning (ML) model and the internists. Developing yet another ML model for sepsis mortality risk is not exciting; but a comparison of the model with human expert internists is exciting. As the key contribution of the paper this comparison has to be solid. Unfortunately, the way the paper stands now, I do not feel that this comparison is rock solid. Here are some weakness in the comparison.(a) Although the difference between internist and ML is high, with only 4 internists, the differences among the internists can significantly influence the results. It would be helpful to have inter-rater agreement among the internists. Also, how was specificity/sensitivity for the ML model determined?

We thank the reviewer for this in-depth analysis of our manuscript. We agree that the differences between the internists could influence the results, this is also highlighted in the individual differences in sensitivity and specificity (S4 Fig). Furthermore, we calculated the inter-rater agreement between internists using Cohen’s Kappa statistics which is described in the table below (in the attached rebuttal document). This information was included in our manuscript as S6 Table, and described in the methods (page 14, lines 254-258) and results section (page 20, lines 340-342).

As an additional sensitivity analysis, we compared the model to alternating physician group compositions (table in the attached rebuttal document)). In each comparison between the machine learning model and the physicians group, a single physician was removed from the physician group. In all comparisons the machine learning model outperforms the physicians. This information was included in our manuscript as S7 Table, and described in the methods (page 14, lines 258-260) and results sections (pages 20-21, lines 342-345). This analysis, supported by the moderate to good Cohen Kappa’s, shows the higher performance of the machine learning model was not due to systemic underperformance of a single physician.

The specificity and sensitivity of the machine learning model was calculated using a standard cut-off of 0.5. We added an additional sentence to the manuscript in the methods section to clarify this: “A machine learning prediction of higher than 0.50 was considered as a positive prediction.” (page 13, lines 228-229).

2. It should also be noted that the machine learned model incorporates physician judgement. Whether a lab test is performed or not is based on clinical judgement and is made available to the model (as the lab absence/presence indicator). A person with fewer labs have fewer problems and is hence less likely to die. How the authors handle missing values is typically completely reasonable and correct, however, in this case, it "leaks" physician knowledge to the ML model.

We agree with the reviewer that the addition of an “absence/presence” indicator also adds a certain level of physician intuition into the laboratory model. We feel however that this is not necessarily problematic from an application perspective. Noteworthy, the most important difference between the laboratory and the laboratory/clinical datasets is the explicit addition of clinical and vital characteristics such as blood pressure, heart rate and oxygen saturation. These factors are known to be important predictors of mortality, which is also observed in our SHAP analysis (Figure 3).

3. Beside the comparison, the model development itself raises some concerns. Specifically, he performance differences between the various machine learning methods is incredibly high (.63-ish for Ridge regression, .65 neural networks, .72 for random forest, but .85 for xgboost). We also typically see xgboost outperform other methods but not to this extreme extent. I am wondering whether the authors may have made some mistake evaluating xgboost. There are other indicators of a possible mistake:

- The performance of xgboost differs across different tables (.813 in Supp Tbl 2; .852 in Supp Tbl 5). With a stated confidence interval of .79-.83, .852 is quite a bit outside the confidence interval.

We share the reviewers’ observation that XGBoost (and other gradient-boosting implementations) often outperform other algorithms in settings with heterogeneous, tabular datasets. As mentioned by the reviewer, we describe the baseline performance of XGBoost in S2 Table with an area-under-the ROC curve of 0.813 (0.791 – 0.835). This was done as a baseline comparison on the development variant of the laboratory dataset (n=1,244) with five-fold cross validation. This model is similar to the “lab” model presented in Figure 2, that has an AUC of 0.82 (0.80-0.84). The minor difference in AUC is most likely due to a different random initialization and different cross-validation folds being created (both models were developed in independent experiments). The XGBoost model we present in S5 Table is the best performing model from Figure 2 which is the laboratory + clinical model. This model had comparable performance with five-fold cross validation in the development dataset (AUC: 0.84, Figure 2) and in the validation dataset (AUC: 0.852, Supplementary Table 5).

To prevent any misinterpretation, we added “on the laboratory dataset” (page 10, line 177) in the manuscript and “using the laboratory dataset” (page 7, line 113) in the supplemental file.

- The learning rate is stated as .075 in Supp Tbl 5 but .001 in the text; could be a typo or actual different hyper-parameterization. A different hyper-parameterization could explain observed AUC values outside the confidence interval than random chance.

We apologize, this was a typo indeed. We corrected the learning rate of 0.001 to 0.075 in S1 supporting information in our supplemental files (page 5, line 97).

- The "95%"-confidence intervals are not 95%. In Supp Fig. 2B, only fold 1 falls consistently within the stated 95% confidence interval, the remaining 4 folds fall outside the "95%"-confidence interval for large consecutive portions of the ROC curve. How was the bootstrap estimation performed? Specifically, which data set is resampled (the development or the leave-out)?

The reviewer is correct: the 95% CI is based on the mean and standard deviation of 5 ROC calculations, each time using 4 of 5 equally sized folds of the development set. Five calculations are however insufficient to assume normality and calculate a reliable 95% CI. One slightly deviating ROC estimation can hence fall –just by statistical chance- almost completely outside the 95% CI. To prevent any confusion, we removed the 95% CI areas and calculations in S1 Fig and Fig 2 (main article) and only present the ROC’s and mean of the 5 individual folds.

4. I agree with the authors that a full-fledged lattice search of the hyper-parameter space is unnecessary, but I would suggest:

- explaining how the hyper-parameters were determined (e.g. using the default values from the package)

Hyperparameter tuning is often critical in the development of high-performance machine learning models but due to the following constraints we decided not to perform any explicit hyperparameter tuning:

* Our main aim is the comparison of machine learning models versus physicians and not the development of a high-performance machine learning model which would be unrealistic given the current sample size

* It would require a third dataset to independently evaluate the different hyperparameters (e.g. a ‘tuning’/’validation’ dataset); which in our case would limit our sample size even further. An alternative would be nested cross-validation (Parvandeh S, et al. Bioinformatics. 2020;36(10):3093-8. doi:10.1093/bioinformatics/btaa046) but given the clinical context of the study we believe this is overly complex.

In the table below (in the attached rebuttal document) we describe the rationale for each of the chosen hyperparameters in our study. This information was added as an additional paragraph in our supplementary information in S3 Table (page 7, line 120-124).

- explaining which if any hyper-parameters were changed (based on the CV test set; NOT the 100 patient leave-out set)

We would like to refer to the comment above; we did not adjust any hyperparameters.

- conduct a sensitivity analysis demonstrating that small changes in the hyper parameters do not lead to major perturbations in the performance.

To illustrate that small changes in hyperparameters do not lead to major pertubations in performance, we performed a grid-search with “max_depth” and “n_estimators” which are known to be amongst the most influential hyperparameters in tree-based models. We compared combinations of these hyperparameters and depicted their AUC in the table below (in the attached rebuttal document). Models employed in our manuscript are highlighted in bold.

We would like to stress that based upon these results we cannot decide which model is better as this would require an additional dataset. Instead, these results only provide a comparison between different hyperparameters, depicting that subtle differences in hyperparameters do not lead to major changes in performance. As the main objective of our study is to compare the machine learning models versus physicians, we would propose not to include this in the current manuscript.

5. The SHAP score is poorly explained and Figure 3 is poorly described. Readers not familiar with SHAP and violin plots will not understand this figure. An example could be useful. Eg. High values of urea (red points) impact the risk of mortality positively (large positive impact); low values reduce it slightly (modest negative impact); urea observations in most patients have no impact on mortality (the urea violin plot is widest at 0 impact).

To facilitate the interpretation of the SHAP algorithm, we added an extended description in the methods section of our paper (pages 6-7, lines 206-216) explaining the interpretation of these values and hereby hopefully facilitating the reader with enough information to interpret these values. Moreover, as the reviewer suggested, we added an example of the interpretation of SHAP values to our results section (page 18, lines 306-309).

6. A concern with xgboost is the (somewhat) black-box nature of the model. The authors address this concern by computing the SHAP score for each feature. Given that the key differentiator of trees from the other methods is their ability to detect and use interactions, SHAP may not capture this well. A better approach is to take the (say) top 20 features and build a reduced model to see the performance loss. (I picked 20 because that is what the authors focus on in Supp Tbl 4, but other numbers could illustrate their point better.)

We agree with the reviewers comment that it would be interesting to build a XGBoost model with the top-N features. We assessed the performance of XGBoost models with top-20, 15, 10, 5 and 3 features in cross-validation by AUC under the ROC curve. Results are depicted in the table below (in the attached rebuttal document). 

The differences between the full and top-20 models are relatively small. Considering the scope of the current manuscript and the small differences, we would propose to not include this into the current manuscript.

7. Clinical significance.ML models in sepsis abound with very marginal contributions. Proposing yet another one without explaining how it can improve sepsis care is fairly meaningless. I appreciate the authors mentioning how their model could be used, and I think expanding on this is essential. The author's suggestion of using the model to identifying high risk patients (>= 50% risk of mortality) is reasonable. However, their evaluation does not quantify their contribution from this perspective. The clinical significance of this work could be significantly improved by (i) showing how much better the ML model is at identifying patients at >=50% risk (or any other high risk); (ii) clinically describing where the ML model was correct and the internists were not (again >= 50% vs <50% can be used) ; (iii) understanding the limits of the model, clinically describing where the ML model "fails": predicts low risk of mortality while the patient dies (regardless of whether the internists made the correct prediction). Note that higher AUC does not guarantee an improved ability to identify high-risk patients; correctly ordering low-risk patients will increase the AUC yet it has not clinical significance.

We agree with the author that the development of machine learning models without any clinical application is meaningless. However, given our limited sample size, only moderate to good model performance and a main focus on the comparison of physician vs machine learning model, we believe that providing these clinical estimates is beyond the scope of the current study. However, in an ongoing follow-up study we build high-performance machine learning models (AUCs of 0.90 and higher) in four Dutch hospitals, including more than 260.000 patients. That study is more focused on the strategy towards clinical application, which would briefly be described as follows:

First, we defined the acceptable percentage of patients that are erroneously identified as “low-risk” by the algorithm (any number from 0-100%). This percentage, e.g. 1%, could be derived from an inventory of acceptable risk tolerance for adverse events by patients, health care workers, or both (Brown TB, et al. J Emerg Med. 2010;39(2):247-52). Then, use the corresponding negative predictive value (in this case 99%) to derive the matching algorithm prediction threshold (e.g. 0.05) and associated values for sensitivity, specificity, and proportion of subjects identified as low risk. A similar approach can be applied to identify high risk patients: define the positive predictive value that would provide an acceptable balance between true high risk patient identification and false positives, e.g. a positive predictive value of 75% would categorize x% as high-risk individuals with 1 in 4 “flaggings” by the clinical decision support tool being false positive. A higher proportion of high risk subject identification is feasible but will be at the expense of increased false positive flaggings.

8. Limitations. Some important limitation are omitted.1. The key limitation is portability. We can expect the risk scores and the internists to have similar performance if they made predictions for patients in a different health system. How about the xgboost model? Will it achieve similar performance?

We expanded on the portability of our model in our discussion section:

(i) “our sample size (especially in the area of machine learning) is relatively small, and therefore substantial further improvement of diagnostic accuracy is likely when increasing the sample size” (page 23-24, lines 406-408)

(ii) “We also limited ourselves to sepsis patients presenting to the ED, and thus it is unknown to what degree these models translate to a broader, general ED population” (page 24, lines 408-410)

(iii) “these results are based on a retrospective, single-center study which limits the external validity of our models” (page 24, lines 410-412)

9. Physicians, which actually seeing patients, may incorporate other features not captured by the EHR.

We agree with the reviewer’s comment that the performance of physicians might be underestimated due to the retrospective nature of the study. We briefly discussed this limitation in our discussion, and extended this with the sentence: “As such, the performance of internal medicine physicians might be underestimated as they were not able to directly see the patient.” (page 24, lines 417-419)

10. SUMMARY. The key contribution of the paper is a comparison between an ML model and human internists. There are minor flaws in the comparison (ML has access to clinical judgement). The model development process appears to have technical problems (confidence intervals appear incorrect; unknown rational for the hyper-parameterization and dependence of the performance on these parameters). The performance envelope of the model is unexplored: what clinical patient characteristics make it fail? make it perform better than the internists? Important limitations are not mentioned. I think this paper has the potential to be an influential piece of work but its contribution as it stands now is insufficient and some technical aspects may even be incorrect.

We again would like to express our gratitude for the thorough review and suggestions of the reviewer. We think that the manuscript significantly improved and hope that it now meets the standard for publication in PLOS ONE.

Reviewer 2

We wish to thank the reviewer for evaluating our manuscript and for providing suggestions and comments to improve our manuscript. The comments have been addressed in a point-by-point fashion in this document. The changes are highlighted in track changes throughout the manuscript. The pages and lines mentioned for each adjustment refer to the manuscript and supplemental file with track changes.

1. Mortality prediction in patients with sepsis, utilising a small dataset of patients presenting to a single emergency department. The title is a little sensational – “Machine Learning versus Physicians”, and could be modified to represent the science “A comparison of machine learning models versus clinical evaluation for mortality prediction in patients with sepsis”

According to the reviewer’s suggestion we changed the title to: “A comparison of machine learning models versus clinical evaluation for mortality prediction in patients with sepsis” (page 1, lines 3-4).

2. Authors testing the hypotehesis that machine learning models would outperform physician evaluation and existing clinical risk scores. Authors determined that machine learning models out performed clinicians due to higher sensitivity and specificity, however discriminatory information is not provided in the abstract.

The discriminatory ability of models versus physicians is described in S5 Table. To further emphasize this, we added this information in our abstract: “The model had higher diagnostic accuracy with an area-under-the-ROC curve of 0.852 (95%CI: 0.783-0.922) compared to abbMEDS (0.631,0.537-0.726), mREMS (0.630,0.535-0.724) and internal medicine physicians (0.735,0.648-0.821).” (page 4, lines 58-61).

3. The references noted in the introduction note high discriminatory scores for abbMEDS (up to 0.85), mREMS (up to 0.84) and clinician judgement (up to 0.81), suggesting similar performance to this cohort

Clinical risk scores and even clinical judgement can indeed show high discriminatory scores (up to 0.80) but vary substantially between centers. We therefore feel it is only scientifically sound to make comparisons within studies. In our study, we found AUC’s of 0.631 (0.537-0.726) and 0.630 (0.535-0.724) for abbMEDS and mREMS, respectively.

4. The comment is made that “machine learning can extract information from incomplete…data” – although complexity and non-linear relationships are areas where ML does succeed, incomplete data is still a major limitation

We agree with the comment of the reviewer that incomplete missing data is still an ongoing limitation in most machine learning models. As complex and non-linear relationships are areas where machine learning currently excels, we adopted reviewer’s comment and modified the sentences in our introduction section accordingly (page 6, line 93).

5. Patients with missing clinical data were excluded – how many variables needed to be missing? Were the data missing at random? A plot representing percentage of missing values would be valuable, and an analysis to confirm that the missing value distribution was equal in the training and testing set.

We included 1,420 patients in our current study of whom 76 were excluded due to missing laboratory results (n=53) or clinical data (n=23). Missing clinical data (n=23) was a result of missing blood pressure (n=5), glasgow coma score (n=7), oxygen saturation (n=6), heart rate (n=4) and age (n=1). We would like to stress that these were excluded before randomization of the dataset into training and testing set. Hence, similar as in a randomized controlled trail (RCT), statistical analysis after randomization to confirm equal distribution would be undesirable.

6. Similarly, the authors note that the machine learning model is capable of dealing with missing data – could further information be provided as to how GBMs deal with the issue? Further, this statement is incongruent with the previous noting that patients with missing clinical data were excluded (unless this does not refer to biomarkers, and instead to other data)

Gradient-boosting systems are models that build an ensemble of ‘weaker’ prediction models, mostly decision trees. Within such decision tree, each node is a test on a feature (e.g. CRP < 5 U/L or age < 70) and each branch represents an outcome for this test. During the training phase of these algorithms, we determine the features (and their cut-offs within a node) that result in the best performance given the dataset. When we specifically deal with a missing value during training, we evaluate what happens with the performance if we take one of the two leaves for the decision tree; the leaf that results in the highest performance increase on average will be defined as the “default” path to take in case there is a missing value. An extended description of this native mechanism for support of missing data in the XGBoost algorithm specifically can be found in its original paper (Chen T, Guestrin C. XGBoost: A Scalable Tree Boosting System. arXiv:1603.02754).

The reviewer is correct that the missing clinical data does not include the biomarkers. We adjusted the manuscript to further clarify this (page 7, line 117).

7. As biochemical markers were taken within the first two hours, and a significant proportion of treatment is performed in that time frame, was the variable of time between presentation and time of biomarker withdrawal recorded and adjusted for?

The reviewer highlights an interesting point about the time of biomarker withdrawal and its relationship to treatment in the emergency department. Unfortunately, we do not have any data on these time windows, but considering that this is evenly distributed in the training and test dataset, it is automatically captured by the model, and will not limit the performance of the models.

8. The train/test split ratio is unusual – 93:7 – can the authors explain why this was chosen? Internal validation on 100 patients is small and may affect the reproducibility of this result.

We agree with the reviewer that the train:test split ratio differs from more conventional splits of e.g. 70%/30%. However, as the main aim of the study is the comparison versus physicians, we decided that 100 patients (which corresponds to 7%) would be a feasible amount of patients to have evaluated by our internal medicine specialists.

9. Mortality was obtained from the electronic health record – was this linked to a national health index? How were deaths out of hospital recorded in the EHR?

Mortality information was acquired through the electronic health records which are linked to municipal population registries.

10. The septic shock definition mentions a MAP ≤90 – is an alternate threshold intended here?

We thank reviewer for this comment as we intended to describe the MAP (mean arterial pressure) here but at a threshold of <65 instead of <90. This was adjusted in the manuscript accordingly (page 10, line 165).

11. The use of SHAP to explain the findings are an important addition and the authors should be credited for its inclusion.

Thank you for the comment. In response the other reviewer, we extended the description of the SHAP algorithm in the methods (pages 6-7, lines 206-216) and results section (page 18, lines 306-309).

12. Why were internal medicine physicians chosen, as opposed to emergency or intensive care physicians? Is this typical for this hospital?

We selected internal medicine physicians (n=4) that were either residents (n=2) or consultants (n=2) specialized in acute internal medicine. At the time of the study, our hospital had no emergency medicines, but rather internal medicine physicians that specialized into acute internal medicine. These physicians work at our emergency department on a day-to-day basis, and thus best suit the comparison versus a machine learning model. To clarify this we modified the manuscript at several lines to explicitly include the word “acute” into our definition of internal medicine specialists (page 13, line 224 and 232; page 20, line 324).

13. Calibration measures such as the Brier score and calibration plots should be presented for the models

We added calibration plots and brier scores in a five-fold cross validation setting of our laboratory and laboratory+clinical models in S2 Fig. This was also added to our manuscript in the methods (page 14, lines 246-247) and results section (page 17, lines 286-287).

14. There is marked class imbalance, with only 13% of the population experiencing the primary outcome of death – were oversampling methods considered?

We thank reviewer for the suggestion. In early stages we indeed considered oversampling but given the positive outcome (death) of 13%, which is reasonably high, we decided not to perform oversampling. Also, oversampling synthesizes and/or duplicates (nearly) identical positive samples from the training dataset, which can lead to potential overfitting especially at smaller sample sizes.

• Results and Discussion

15. There appears to be a significantly higher percentage of patients with cancer and diabetes in the training/development set

Similar to other randomization procedures (e.g. in a RCT), statistical chance may lead to slight imbalances between randomized groups. For critical parameters one can choose to distribute them evenly during randomization, but that was considered unnecessary in our study.

16. Why is the discrimination/AUC not directly compared between the ML models and the physicians and clinical scores?

The analysis of discrimination for the physicians versus models is described in S5 Table. We deliberately chose not to use discrimination (AUC under the ROC) as the main analysis in our manuscript as it might not be fully appropriate for the evaluation of the performance of internal medicine physicians. Typically, a ROC analysis shows how sensitivity (true positive rate) changes with varying specificity (true negative rate or 1 −false positive rate) for different thresholds. Internal medicine physicians, however, only provide a binary prediction (death or survive) and therefore will represented by a single point in the ROC analysis. Although this still allows calculation of an area-under-the curve, it is an ongoing debate whether or not this is a valid metric (Muschelli J. ROC and AUC with a Binary Predictor: a Potentially Misleading Metric. Journal of Classification. 2019. doi:10.1007/s00357-019-09345-1).

Nonetheless, we performed a ROC analysis and depicted the predictions of internal medicine physicians as bullets in the figure (see figure below in the attached rebuttal document). This figure is also embedded as S3 Fig in our manuscript (page 20, line 337) and supplemental files (page 12, lines 172-177).

Furthermore, we recognize that the discriminatory ability of diagnostic tool is widely used and thus we decided to explicitly add a sentence regarding this comparison in our results section: “Additionally, the model had higher overall diagnostic accuracy as depicted by an AUC of 0.852 (95% CI: 0.783-0.922) compared to abbMEDS (0.631, 0.537-0.726), mREMS (0.630, 0.535-0.724) and internal medicine physicians (0.735, 0.648-0.821).” (page 20, lines 335-338).

17. AUCs should be compared using DeLong’s test to demonstrate statistical superiority in regard to discrimination

We thank reviewer for the suggestion of adding DeLong’s test to test for statistical significance in our ROC analysis. However, we feel that calculating these for binary predictors could be misleading. DeLong’s test is mainly useful for testing differences between related predictors (e.g. an old risk score versus an updated risk score with additional variables), which is not the case in the current study.

18. Figure 2 does not include the clinical risk scores for comparison

Figure 2 shows the development of machine learning models using the development dataset. In our validation dataset of 100 patients, we provide a head-to-head comparison of the clinical risk scores with our machine learning models and internal medicine specialists (Figure 4 and S5 Table).

19. Creatinin does not share the same relationship with model output as does urea – can this be explained clinically? (from the SHAP figure)

From Figure 3 we can indeed observe that urea and creatinin have different relationships with the predictions. Urea contributes to a more positive prediction (death) especially at higher levels (indicated by the red dots), whereas creatinin mainly has a negative (protective) effect at lower biomarkers levels (indicated by blue dots). Creatinin is a marker for kidney function (and slightly for muscle mass), whereas urea is also a marker for kidney function but moreover is highly associated with hemodynamics. Hence, urea is an important marker for the overall disease state of a patient. This is also reflected in recent risk scores, such as the RISE UP, which include urea but not creatinin in their prediction models (Zelis N, Buijs J, de Leeuw PW, van Kuijk SMJ, Stassen PM. Eur J Intern Med. 2020;77:36-43.doi:10.1016/j.ejim.2020.02.021).

20. Several references to other machine learning models in the literature focused on sepsis and mortality prediction have been omitted

Based on the comment of the reviewer we conducted a new literature search and found four additional references that were included in the revised version of our manuscript:

* Horng S, Sontag DA, Halpern Y, Jernite Y, Shapiro NI, Nathanson LA. Creating an automated trigger for sepsis clinical decision support at emergency department triage using machine learning. PLoS One. 2017;12(4):e0174708. doi:10.1371/journal.pone.0174708

* Ford DW, Goodwin AJ, Simpson AN, Johnson E, Nadig N, Simpson KN. A Severe Sepsis Mortality Prediction Model and Score for Use With Administrative Data. Crit Care Med. 2016;44(2):319-27. doi:10.1097/CCM.0000000000001392

* Shukeri W, Ralib AM, Abdulah NZ, Mat-Nor MB. Sepsis mortality score for the prediction of mortality in septic patients. J Crit Care. 2018;43:163-8. doi:10.1016/j.jcrc.2017.09.009

* Bogle B, Balduino, Wolk D, Farag H, Kethireddy, Chatterjee, et al. Predicting Mortality of Sepsis Patients in a Multi-Site Healthcare System using Supervised Machine Learning2019.

21. Funded study with no conflicts of interest

Our study was funded by the non-profit organization Dutch Federation of Clinical Chemistry (NVKC). As stated in our disclosure the funders had no role in the study design, data collection, analysis, decision to publish, and preparation of the manuscript.

22. Grammar could be slightly improved however does not interfere with message of the paper

o “…and categorize from low to high risk” should be “and is categorized from low to high risk”

o “follow-studies” should be “follow-up studies”

We carefully re-read the manuscript and performed several minor modifications to improve the readability and grammar of our manuscript.

23. Authors have made data available without restriction, however note that it is available in the supplementary material – the code and raw data are not provided?

Our supplementary material presents additional information regarding the manuscript. On reasonable request the raw data and the source code used in the current study are available.

---

## [Decision Letter · Decision Letter 1]

26 Aug 2020

PONE-D-20-09068R1

A comparison of machine learning models versus clinical evaluation for mortality prediction in patients with sepsis

PLOS ONE

Dear Dr. Meex,

Thank you for submitting your manuscript to PLOS ONE. After careful consideration, we feel that it has merit but does not fully meet PLOS ONE’s publication criteria as it currently stands. Therefore, we invite you to submit a revised version of the manuscript that addresses the points raised during the review process.

In particular, please consider Reviewer 2's concerns regarding size of validation subset. I understand the authors point on attempting the fairest possible comparison between ML algorithms and clinicians, but also agree that a very small validation subset could be in detriment of the quality of the ML performance evaluation. The authors might want to consider the inclusion of further model evaluation using an out-of-the-bag resampling strategy on 70/30 as suggested, alongside the ones already presented. It would be very helpful if you could address this and the rest of the reviewer points.

We look forward to receiving your revised manuscript.

Kind regards,

Ivan Olier, Ph.D.

Academic Editor

PLOS ONE

Reviewers' comments:

Reviewer's Responses to Questions

**Comments to the Author**

1. If the authors have adequately addressed your comments raised in a previous round of review and you feel that this manuscript is now acceptable for publication, you may indicate that here to bypass the “Comments to the Author” section, enter your conflict of interest statement in the “Confidential to Editor” section, and submit your "Accept" recommendation.

Reviewer #1: All comments have been addressed

Reviewer #2: All comments have been addressed

2. Is the manuscript technically sound, and do the data support the conclusions?

Reviewer #1: Yes

Reviewer #2: Partly

3. Has the statistical analysis been performed appropriately and rigorously? 

Reviewer #1: Yes

Reviewer #2: No

4. Have the authors made all data underlying the findings in their manuscript fully available?

Reviewer #1: Yes

Reviewer #2: No

5. Is the manuscript presented in an intelligible fashion and written in standard English?

Reviewer #1: Yes

Reviewer #2: Yes

6. Review Comments to the Author

Reviewer #1: (No Response)

Reviewer #2: Reviewer comments are attached below:

- The significantly lower discriminatory scores in this data compared with the literature may represent differences in the particular sample, or reflect the small sample size and thus be inaccurately portraying a lower discriminatory capacity of the physicians.

- The decision to only test on 100 patients is unclear; the model may falsely appear more accurate without robust validation. Did the authors attempt validation on a randomly selected 30%?

- Cancer and diabetes would traditionally be strongly associated with mortality, particularly in septic shock. The authors note that this was considered unnecessary to distribute these evenly, however it is unclear why.

- A statistical opinion should be sought regarding the direct validity of comparing AUROCs with DeLong’s test; it is the reviewers opinion that this is a useful comparator for discriminatory evaluations such as this.

- The reverse relationships of urea and creatinine are unclear. Often both are not included in the same score as they are in the same direction (and consequently one will knock out the other during development); it remains unclear why they would be in opposite directions.

7. PLOS authors have the option to publish the peer review history of their article (what does this mean?). If published, this will include your full peer review and any attached files.

Reviewer #1: No

Reviewer #2: No

---

## [Author Response · Author response to Decision Letter 1]

1 Sep 2020

Reviewer 2

We want to thank the reviewer for re-evaluating our manuscript. The comments have been addressed in a point-by-point fashion in this document. The changes are highlighted in track changes throughout the manuscript. The pages and lines mentioned for each adjustment refer to the manuscript and supplemental file with track changes.

1. The significantly lower discriminatory scores in this data compared with the literature may represent differences in the particular sample, or reflect the small sample size and thus be inaccurately portraying a lower discriminatory capacity of the physicians.

Clinical risk scores show varying discriminatory performance in the literature with area-under-the receiver operating characteristic (AUROC) ranging between 0.62-0.85 for abbMEDS and 0.62-0.84 for mREMS. The AUC in in our study for both scores is within the range reported in literature. 

2. The decision to only test on 100 patients is unclear; the model may falsely appear more accurate without robust validation. Did the authors attempt validation on a randomly selected 30%? 

We would like to emphasize that these 100 patients represent a random selection from the population. The validation number of 100 patients was chosen as an amount that was feasible to have carefully evaluated by 4 physicians. In regard to the reviewers suggestion to attempt validation on a randomly selected 30%, we think there may be a misunderstanding which we try to clarify: in our analysis we perform model evaluation on a randomly selected 20%, and then on top applied 5-fold cross validation, each time with another random 20% selection. We feel that our approach with 5-fold cross validation is in fact more robust than a single validation with 30%. Nevertheless, we carried out the validation suggested by the reviewer where we used a randomly selected subset of 70% to train laboratory and laboratory + clinical models, and a randomly selected 30% to evaluate the model. The AUC’s (depicted below in Response To Reviewers document) of the resulting models are 0.84 and 0.86, which is comparable to the performance reported in our manuscript (0.82 [0.80-0.84] and 0.84 [0.81-0.87] respectively). 

We feel that adding 70/30% resampling strategy on top of the 80/20% 5-fold cross validation to our current manuscript, would be redundant and may even be confusing to the journal’s readership. We therefore propose not to include this additional analysis in the manuscript. Alternatively we will be happy to use the possibility offered by the journal to make this review correspondence publicly available alongside the article, so interested readers are able to see the additional analysis and read all considerations made during the review process.

3. Cancer and diabetes would traditionally be strongly associated with mortality, particularly in septic shock. The authors note that this was considered unnecessary to distribute these evenly, however it is unclear why.

This choice is similar to decisions in a randomization procedure in a randomized controlled trial: statistical chance may lead to slight imbalances between randomized groups (which becomes less likely with increasing sample size). If there is a single (or very limited number) of key confounding factors, one can choose to account for them and guarantee an equal distribution. In our study however, we feel that an even distribution of cancer and diabetes is not substantially more critical than many other traits: e.g. age, sex, hemodynamics at presentation, other comorbidities etc. In the absence of compelling a priori evidence to prioritize cancer and diabetes over all other potentially confounding parameters, we chose not distribute them evenly. 

4. A statistical opinion should be sought regarding the direct validity of comparing AUROCs with DeLong’s test; it is the reviewers opinion that this is a useful comparator for discriminatory evaluations such as this.

As suggested by the reviewer, we compared the discriminatory performance of the machine learning model versus the clinical risk scores and physicians with DeLong’s test (DeLong et al, 1988) in the validation subset. The results are provided in the table below. 

Model AUC (95% CI) P-Value

Machine learning model 0.852 (0.783 – 0.922) N/A

abbMEDS 0.631 (0.537 – 0.726) 0.021

mREMS 0.630 (0.535 – 0.724) 0.016

Internal medicine physicians 0.735 (0.648 – 0.821) 0.189; 0.072; 0.068; 0.032a

a. Individual P-values were calculated for each of the internal medicine physicians. 

We updated the manuscript in the methods section on page 11, lines 230-231 and in the results section on page 15, lines 304-306. Additionally, we updated Supplementary Table 5 on page 8, lines 124-126 with these results. 

5. The reverse relationships of urea and creatinine are unclear. Often both are not included in the same score as they are in the same direction (and consequently one will knock out the other during development); it remains unclear why they would be in opposite directions.

From a clinical perspective creatinin is a marker for kidney function (and slightly for muscle mass), whereas urea also reflects hemodynamics. Urea is therefore an important marker for the overall disease state of a patient. Although creatinin and urea are concordant in many subjects, they can differ, and reverse relationships are actually possible and should not be considered surprising. Indicative for the fact that urea and creatinin are not always concordant is the clinical use of an urea-to-creatinin ratio, which can aid in the diagnosis of prerenal injury, GI bleeding, elderly patients or hypercatabolic states (Irwin & Rippe, 2008; Brisco et al, 2013; Sunjino et al, 2019).

---

## [Decision Letter · Decision Letter 2]

21 Oct 2020

PONE-D-20-09068R2

A comparison of machine learning models versus clinical evaluation for mortality prediction in patients with sepsis

PLOS ONE

Dear Dr. Meex,

Thank you for submitting your manuscript to PLOS ONE. After careful consideration, we feel that it has merit but does not fully meet PLOS ONE’s publication criteria as it currently stands. Therefore, we invite you to submit a revised version of the manuscript that addresses the points raised during the review process.

We look forward to receiving your revised manuscript.

Kind regards,

Ivan Olier, Ph.D.

Academic Editor

PLOS ONE

Reviewers' comments:

Reviewer's Responses to Questions

**Comments to the Author**

1. If the authors have adequately addressed your comments raised in a previous round of review and you feel that this manuscript is now acceptable for publication, you may indicate that here to bypass the “Comments to the Author” section, enter your conflict of interest statement in the “Confidential to Editor” section, and submit your "Accept" recommendation.

Reviewer #3: All comments have been addressed

Reviewer #4: (No Response)

2. Is the manuscript technically sound, and do the data support the conclusions?

Reviewer #3: Yes

Reviewer #4: Partly

3. Has the statistical analysis been performed appropriately and rigorously? 

Reviewer #3: I Don't Know

Reviewer #4: I Don't Know

4. Have the authors made all data underlying the findings in their manuscript fully available?

Reviewer #3: Yes

Reviewer #4: Yes

5. Is the manuscript presented in an intelligible fashion and written in standard English?

Reviewer #3: Yes

Reviewer #4: Yes

6. Review Comments to the Author

Reviewer #3: I have no further comments for the authors. I have read the prior comments and believe that they have beed addressed in a satisfactory manner.

Reviewer #4: Thank you for the opportunity to review this manuscript. Though interesting and seemingly sound in terms of methods, I have some concerns regarding its application (see comments below).

In order for this study to be valid it needs to applied to an undifferentiated population with infection who could have sepsis, but do not have the diagnosis yet. As I read it, it would appear that all of the patients in this study were referred for admission for some reason, which is different from patients presenting to the ED with an infection as many of those patients would be discharged home. Therefore I don’t believe this score can be directly compared to score made for an undifferentiated ED population. However, if this is not the case, the authors should state clearly that this study included all ED data for patients meeting their SIRS/qSOFA criteria.

I would be interested in knowing how this score would be applied, since the authors specifically chose to test and validate their models based on the first 2 hours of available clinical and laboratory information and from what I can tell these were all patients who were being admitted to the hospital. If this is the case it would reduce the applicability of this tool.

How was consent obtained on a retrospective study? How were patients able to refuse participation, since the “ethics committee waived the requirement for informed consent”

This article should have a separate stat/methods review, particularly the use of 100 patients for the validation set despite this being addressed in this revision in a 70/30 split the data in table 1 seem too sparse with several of the features having N’s in the single digits. I am not sure the standard for an ML paper. Also I am not familiar with their method for cross-validation.

I am not familiar with the term acute internal medicine physicians? Do these physicians work in the ED or do they work on the acute inpatient services admitting patients?

How was the subpopulation of 1420 patients selected out of the 5967 patients consulted to internal medicine? Was this cohort selected randomly? There should be flow diagram showing the total populations, those excluded with reasons why and the final cohort.

The lab model vs the lab + clinical model include features that are quite different. How do the authors suggest we reconcile these differences and which model should we consider to be superior? Also please define “Blood group (present)”.

Some terms are not defined in the manuscript, such as GCS. Also thrombocytes should be replaced with “platelet count”.

The standard for mortality prediction in sepsis is the SOFA score. There should be a direct comparison with SOFA or at least modified version of the SOFA score (there are several versions) in order to conclude that this may have clinical utility.

There needs to be more of an explanation of the different models and how they should be interpreted.

7. PLOS authors have the option to publish the peer review history of their article (what does this mean?). If published, this will include your full peer review and any attached files.

Reviewer #3: No

Reviewer #4: No

---

## [Author Response · Author response to Decision Letter 2]

5 Nov 2020

Reviewer #3

1. I have no further comments for the authors. I have read the prior comments and believe that they have been addressed in a satisfactory manner.

Response: We wish to thank the reviewer for evaluating our manuscript. 

Reviewer #4

1. Thank you for the opportunity to review this manuscript. Though interesting and seemingly sound in terms of methods, I have some concerns regarding its application (see comments below). 

Response: We want to thank the reviewer for evaluating our manuscript and for giving important suggestions and comments to improve our manuscript. The comments have been addressed in a point-by-point fashion in this document. The changes are highlighted in track changes throughout the manuscript. The pages and lines mentioned for each adjustment refer to the manuscript and supplemental file version with track changes. 

2. In order for this study to be valid it needs to applied to an undifferentiated population with infection who could have sepsis, but do not have the diagnosis yet. As I read it, it would appear that all of the patients in this study were referred for admission for some reason, which is different from patients presenting to the ED with an infection as many of those patients would be discharged home. Therefore I don’t believe this score can be directly compared to score made for an undifferentiated ED population. However, if this is not the case, the authors should state clearly that this study included all ED data for patients meeting their SIRS/qSOFA criteria.

Response: This study focused on sepsis patients who visited the ED. All patients aged ≥18 years being referred to the internal medicine physician because of sepsis (i.e. a suspected or proven infection with two or more SIRS and/or qSOFA criteria) (S1 supporting information) were included in this study. This is also described in our methods section at page 6, lines 99-109. Our machine learning model was compared to the abbreviated Mortality in Emergency Department Sepsis (abbMEDS) and the modified Rapid Emergency Medicine Score (mREMS). Since the abbMEDS is specifically designed for sepsis patients, we believe this is a fair comparison. We agree with the reviewer that mREMS is a score originally developed for an undifferentiated ED population, but since it has also been validated specifically in sepsis populations (Chen et al., 2013, BMJ Emer Med J; Howell et al., 2008, Acad Emer Med; Sankoff et al., 2008, Crit Care Med; Crowe et al., 2010, J Emerg Trauma Shock), we believe it is also insightful in a comparison in our study. For completeness, and in response to your request in question 10, we also included the SOFA score in our comparison to the machine learning model and internal medicine physicians. This is described in greater detail in our response to question 10. 

3. I would be interested in knowing how this score would be applied, since the authors specifically chose to test and validate their models based on the first 2 hours of available clinical and laboratory information and from what I can tell these were all patients who were being admitted to the hospital. If this is the case it would reduce the applicability of this tool.

Response: We agree with the author that the development of machine learning models without any clinical application is meaningless. However, given our limited sample size, only moderate to good model performance and a main focus on the comparison of physician vs machine learning model, we believe that providing these clinical estimates is beyond the scope of the current proof-of-concept study. However, in an ongoing follow-up study we built high-performance machine learning models (AUCs of 0.90 and higher) in four Dutch hospitals, including more than 260.000 patients. That study is more focused on the strategy towards clinical application, which can be described as follows: 

First, we defined the acceptable percentage of patients that are erroneously identified as “low-risk” by the algorithm (any number from 0-100%). This percentage, e.g. 1%, could be derived from an inventory of acceptable risk tolerance for adverse events by patients, health care workers, or both (Brown TB, et al. J Emerg Med. 2010;39(2):247-52). Then, we will use the corresponding negative predictive value (in this case 99%) to derive the matching algorithm prediction threshold (e.g. 0.05) and associated values for sensitivity, specificity, and proportion of subjects identified as low risk. A similar approach can be applied to identify high risk patients: define the positive predictive value that would provide an acceptable balance between true high risk patient identification and false positives, e.g. a positive predictive value of 75% would categorize x% as high-risk individuals with 1 in 4 “flaggings” by the clinical decision support tool being false positive. A higher proportion of high risk subject identification is feasible but will be at the expense of increased false positive flaggings. 

4. How was consent obtained on a retrospective study? How were patients able to refuse participation, since the “ethics committee waived the requirement for informed consent”

Response: The study was approved by the medical ethical committee (METC 2019-1044) and hospital board of the Maastricht University Medical Centre+. The ethics committee waived the requirement for informed consent. Patients who are treated in The Netherlands automatically consent for their data to be used for anonymized scientific research provided that medical ethical approval is obtained, unless the specifically refuse this. 

5. This article should have a separate stat/methods review, particularly the use of 100 patients for the validation set despite this being addressed in this revision in a 70/30 split the data in table 1 seem too sparse with several of the features having N’s in the single digits. I am not sure the standard for an ML paper. Also I am not familiar with their method for cross-validation.

Response: The methods and statistical analysis of our manuscript have extensively been reviewed and, after revisions, considered robust by previous reviewers (#2 and #3). Cross-validation is a model validation technique for assessing how the results of a statistical analysis will generalize to an independent data set. This was described in our methods section (page 10, lines 173-183) and is widely used and adapted in literature validating clinical prediction models (few examples include Beker, et al., 2020, Nature Mach Intell; de Rooij et al., 2020, Adv in Meth and Pract in Psych Sci; Saeb et al, 2017, Gigascience; Steyerberg et al, 2014, Eur Heart J).

6. I am not familiar with the term acute internal medicine physicians? Do these physicians work in the ED or do they work on the acute inpatient services admitting patients?

Response: We selected internal medicine physicians (n=4) who were either residents (n=2) or consultants (n=2) specialized in acute internal medicine. At the time of the study, our hospital had no emergency physicians, but rather internal medicine physicians who work at our emergency department on a day-to-day basis, and thus represent the most experienced emergency medicine physicians for a comparison versus a machine learning model. They treat patients at the acute admission unit as well and are familiar with treating sepsis, infections being the main reason for ED visits they handle.

7. How was the subpopulation of 1420 patients selected out of the 5967 patients consulted to internal medicine? Was this cohort selected randomly? There should be flow diagram showing the total populations, those excluded with reasons why and the final cohort.

Response: During the study period 5,967 patients that presented to our emergency department were referred to an internal medicine physician. Of these patients, 1420 patients had a suspected or proven infection and fulfilled two or more SIRS and/or qSOFA criteria. To clarify the complete process from study inclusion to data processing we depicted the flow chart below (in response to reviewers document). This flow chart was also added as S1 Fig (page 10, lines 154-159) and inserted into our manuscript at page 13, line 250. 

8. The lab model vs the lab + clinical model include features that are quite different. How do the authors suggest we reconcile these differences and which model should we consider to be superior? Also please define “Blood group (present)”.

Response: The lab and lab + clinical model include features that are described in S1 Table. Briefly, the lab + clinical model consists of all features in the lab model with additional clinical variables. Figure 3 presents an analysis of the top-20 most important features on a model level using the SHapley Additive exPlanations (SHAP) algorithm (Lundberg et al., 2020, Nat Mach Intell). Hence, all laboratory features ranked in the top-20 lab + clinical model are also important in the lab model. The main difference is that several clinical features (e.g. heart rate, oxygen saturation and systolic blood pressure) appear to be important and therefore are ranked amongst the top-20 features. 

In the current comparison, the lab + clinical model slightly outperformed the lab model (AUC 0.84 vs 0.82) and can therefore be considered superior in terms of performance. In a follow-up study (answer #3) we however decided to continue with the laboratory model as automatic, standardized collection of clinical variables is more complex and subject to variability. 

We apologize for the unclear definition in relation to “Blood group (present)”. This variable represents whether or not the attending physician ordered a blood group assessment of the patient. As such this lab parameter reflects the physicians consideration to request a blood transfusion. We adjusted this in our manuscript in a revised version of Figure 3 and on page 7, lines 131-132 in the supplemental information.

9. Some terms are not defined in the manuscript, such as GCS. Also thrombocytes should be replaced with “platelet count”.

Response: We thank reviewer for the suggestion. To clarify our manuscript we defined several terms in our manuscript, including platelet count (page 15, line 279 in manuscript; page 5, line 106 and page 7, lines 107 in supplementals; revised Figure 2), glasgow coma score (GCS; page 15, line 280) and C-reactive protein (CRP; page 15, line 285-286).

10. The standard for mortality prediction in sepsis is the SOFA score. There should be a direct comparison with SOFA or at least modified version of the SOFA score (there are several versions) in order to conclude that this may have clinical utility.

Response: We thank the reviewer for the suggestion to include the SOFA score in the comparison we provide with the abbMEDS and mREMS clinical risk scores (Figure 4 in manuscript). We decided to implement the SOFA score in our manuscript in its original version (Vincent et al., 1996, Intensive Care Medicine) scoring 1 to 4 points for each of the six organ systems. This was included in our introduction (page 4, line 75; page 5, line 97), methods (page 10, lines 213-214), results (page 13, line 254; page 13, line 258; page 15, line 299; page 16, lines 305-308, 308-309, 312-313, 328-331), discussion (page 18, line 346) and supplemental information (page 3, lines 65-69; S5 Table; S4 Fig). The comparison with the new SOFA score was updated in a revised version of Figure 4 (depicted in the response to reviewers document and manuscript). 

Although it performs better than the other clinical risk scores, the machine learning models still outperforms all three clinical risk scores. We therefore believe it does not impact the major findings of our current manuscript. 

11. There needs to be more of an explanation of the different models and how they should be interpreted.

Response: We assume the reviewer is referring to the different machine learning models explored in the methods section of our manuscript (page 9, line 160-164). The models and their implementation details were described in S2 supporting information (page 4, lines 70-101). The results of the comparison of different models was described in S3 Table. To further clarify our models, we extended the description of our models in S2 supporting information (page 4, lines 70-101).

---

## [Decision Letter · Decision Letter 3]

23 Dec 2020

A comparison of machine learning models versus clinical evaluation for mortality prediction in patients with sepsis

PONE-D-20-09068R3

Dear Dr. Meex,

We’re pleased to inform you that your manuscript has been judged scientifically suitable for publication and will be formally accepted for publication once it meets all outstanding technical requirements.

Kind regards,

Ivan Olier, Ph.D.

Academic Editor

PLOS ONE

Additional Editor Comments (optional):

Reviewers' comments:

Reviewer's Responses to Questions

**Comments to the Author**

1. If the authors have adequately addressed your comments raised in a previous round of review and you feel that this manuscript is now acceptable for publication, you may indicate that here to bypass the “Comments to the Author” section, enter your conflict of interest statement in the “Confidential to Editor” section, and submit your "Accept" recommendation.

Reviewer #4: All comments have been addressed

2. Is the manuscript technically sound, and do the data support the conclusions?

Reviewer #4: Partly

3. Has the statistical analysis been performed appropriately and rigorously? 

Reviewer #4: Yes

4. Have the authors made all data underlying the findings in their manuscript fully available?

Reviewer #4: Yes

5. Is the manuscript presented in an intelligible fashion and written in standard English?

Reviewer #4: Yes

6. Review Comments to the Author

Reviewer #4: The manuscript is substantially improved and I thank the authors for their efforts. However, the main limitation is still present. Namely, that the study was performed in a population that was already consulted for admission to the hospital for sepsis. I am not sure how clinically this score would be applied in the ED setting.

7. PLOS authors have the option to publish the peer review history of their article (what does this mean?). If published, this will include your full peer review and any attached files.

Reviewer #4: No

---

## [Editor Report · Acceptance letter]

5 Jan 2021

PONE-D-20-09068R3 

A comparison of machine learning models versus clinical evaluation for mortality prediction in patients with sepsis 

Dear Dr. Meex:

I'm pleased to inform you that your manuscript has been deemed suitable for publication in PLOS ONE. Congratulations! Your manuscript is now with our production department. 

Kind regards, 

on behalf of

Dr. Ivan Olier 

Academic Editor

PLOS ONE